# Benchmarking Large Language Models with Integer Sequence Generation Tasks

**Daniel O'Malley**
Earth and Environmental Sciences Division
Los Alamos National Laboratory
omalled@lanl.gov

**Manish Bhattarai**
Theoretical Division
Los Alamos National Laboratory
ceodspspectrum@lanl.gov

**Nishath Rajiv Ranasinghe**
Earth and Environmental Sciences Division
Los Alamos National Laboratory
ranasinghe@lanl.gov

**Erick Draayer**
Computational Sciences Division
Los Alamos National Laboratory
edraayer@lanl.gov

**Javier E. Santos**
Earth and Environmental Sciences Division
Los Alamos National Laboratory
jesantos@lanl.gov

## Abstract

We present a novel benchmark designed to rigorously evaluate the capabilities of large language models (LLMs) in mathematical reasoning and algorithmic code synthesis tasks. The benchmark comprises integer sequence generation tasks sourced from the Online Encyclopedia of Integer Sequences (OEIS), testing LLMs' abilities to accurately and efficiently generate Python code to compute these sequences without using lookup tables. Our comprehensive evaluation includes leading models from OpenAI (including the specialized reasoning-focused o-series), Anthropic, Meta, and Google across a carefully selected set of 1000 OEIS sequences categorized as "easy" or "hard." Half of these sequences are classical sequences from the early days of OEIS and half were recently added to avoid contamination with the models' training data. To prevent models from exploiting memorized sequence values, we introduce an automated cheating detection mechanism that flags usage of lookup tables, validated by comparison with human expert evaluations. Experimental results demonstrate that reasoning-specialized models (o3, o3-mini, o4-mini from OpenAI, and Gemini 2.5-pro from Google) achieve substantial improvements in accuracy over non-reasoning models, especially on more complex tasks. However, overall model performance on the hard sequences is poor, highlighting persistent challenges in algorithmic reasoning. Our benchmark provides important insights into the strengths and limitations of state-of-the-art LLMs, particularly emphasizing the necessity for further advancements to reliably solve complex mathematical reasoning tasks algorithmically.

## 1 Introduction

Benchmarking plays a crucial role in the development and evaluation of large language models (LLMs), helping gauge their abilities across various domains such as natural language understanding, knowledge retrieval, and mathematical reasoning. The progress that LLMs have made on challenging benchmarks is remarkable – matching even the performance of expert humans on advanced problems

39th Conference on Neural Information Processing Systems (NeurIPS 2025) Track on Datasets and Benchmarks.

in the human's domain of expertise. With the release of more powerful reasoning models, there is a need for benchmarks that can rigorously test more advanced abilities of these systems.

In this paper, we introduce a novel benchmark based on integer sequence generation tasks sourced from the Online Encyclopedia of Integer Sequences (OEIS) [12, 9]. During evaluation, the model is provided only the OEIS Name and Comments fields (no sequence values, formulas, or OEIS code). Held-out sequence values are used solely for unit tests. The difficulty of these tasks ranges from trivial (A000004 is the sequence of all zeros) to extremely difficult and interesting (A000001 is the number of groups of order n – "a fascinating function" for which Conway et al. [6] recently provided only an approximation of the series). The benchmark therefore spans problems an undergraduate could solve—such as listing the non-square numbers (A000037) through PhD-level research challenges like counting groups of order n (A000001), and Ramsey numbers (A000789, A000791 and A003323), about which Paul Erdös famously said that for the Ramsey number $R(6, 6)$ humanity would have a better chance of destroying an invading alien force than computing it. These tasks are particularly challenging for LLMs, as they require the models not only to understand the sequences but also to implement efficient algorithms that can run quickly (both to test the algorithms and for the expediency of the benchmark). This makes integer sequence generation an excellent testbed, especially for reasoning models, which are optimized for tasks such as mathematics and coding.

Our benchmark consists of a diverse set of 1000 integer sequences labeled "easy" and "hard" in OEIS. We evaluate a wide range of models on this benchmark including reasoning and non-reasoning frontier models. The codes are subject to a time limit that is allowed to vary (similar to a pass@k metric, where different values of k are used), analyzing their performance in terms of both accuracy and efficiency. While reasoning models generally outperform the non-reasoning models, they still struggle, especially with the hard sequences. Additionally, we introduce mechanisms for detecting and preventing the use of lookup tables to verify that models write legitimate code rather than relying on a memorized version of the sequence entries. See Figure 2 for an overview of our approach.

Our contributions are as follows: (1) We introduce a new benchmark for LLMs based on integer sequence generation, emphasizing mathematical and computational reasoning and efficiency. (2) We evaluate numerous frontier LLMs, demonstrating their strengths and limitations in handling these algorithmic tasks. (3) We provide a framework for detecting and mitigating the use of lookup tables in sequence generation tasks, bolstering the integrity of the evaluation process.

## 2   Related Work

Benchmarking has been essential in evaluating the capabilities of LLMs across various domains, particularly in mathematical reasoning and code generation. Existing benchmarks such as MATH [8], GSM8K [5], and HumanEval [4] assess models on complex problem-solving and programming tasks. While these benchmarks provide valuable insights, they often either cover broad problem areas or focus on specific aspects like functional correctness in code generation [1, 2, 3, 4]. Our benchmark distinguishes itself by concentrating on algorithmic reasoning through integer sequence generation, demanding both mathematical insight and efficient code implementation. This approach enables a deeper evaluation of LLMs' capacity to generate accurate, efficient algorithms, filling a gap in existing benchmarks by challenging the latest, most advanced models in a meaningful way.

FrontierMath [7] is a recently released benchmark designed to assess advanced mathematical reasoning in large language models using hundreds of difficult, research-level math problems curated by expert mathematicians. While FrontierMath evaluates deep mathematical insight, it is not designed to be community-maintained and depends heavily on contributions from a small set of invited mathematicians, including Fields Medalists. In contrast, the source of our benchmark test suite is actively maintained by a broader community, with new problems continuously submitted, reviewed, and published, ensuring that the test set is always beyond the training data of any released LLM. Furthermore, whereas FrontierMath problems are all solvable by humans by design (often requiring hours of expert effort), our benchmark includes sequences for which no efficient human-generated solution is known, enabling it to serve as a more ambitious testbed for super-intelligent reasoning capabilities. Finally, our benchmark emphasizes not only deep mathematical understanding but also robust algorithm design: many sequences require numerically stable and computationally efficient implementations. Naive or poorly written solutions may cause underflow, overflow, or fail to complete within tight runtime constraints.

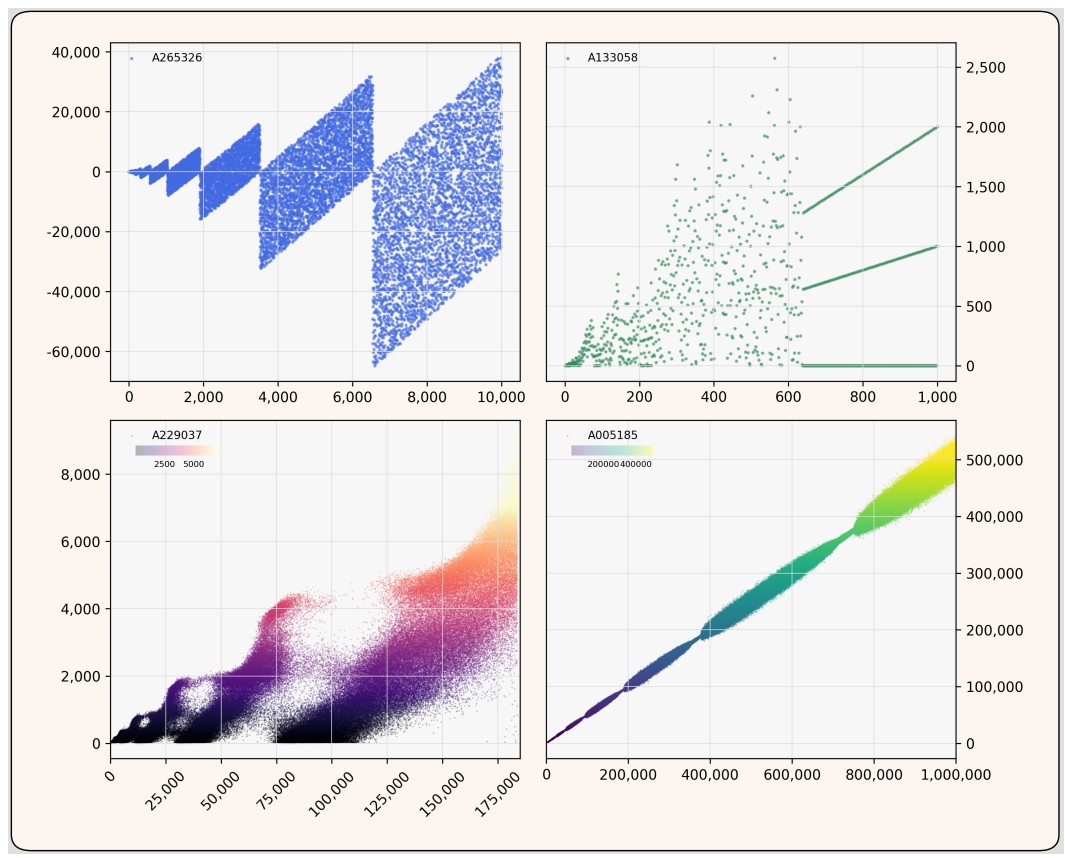

Figure 1: Each panel visualizes an individual OEIS sequence using integer-valued $(n, a(n))$ pairs plotted as raw scatter plots without smoothing. **Top-left (A265326):** This sequence forms a striking pattern of diagonal parallelograms, caused by taking each prime $p$, reversing its binary expansion, and subtracting: $a(n) = p_n - reverse(p_n)$, where $p_n$ is the $n$-th prime. The symmetry arises because reversals often yield other primes, and transitions occur at binary boundaries (e.g., powers of 2), expanding with scale. **Top-right (A133058):** This chaotic-looking trajectory dramatically stabilizes after n=640, where it enters a perfectly repeating three-term loop. N. J. A. Sloane famously compared this to the scene in Avatar where Jake Sully finally tames his Banshee: "fly straight, dammit." **Bottom-left (A229037):** A non-averaging, fractal-like sequence that forbids 3-term arithmetic progressions. Its dense layering and soft envelope illustrate global constraints emerging from a purely local rule. **Bottom-right (A005185):** Hofstadter's Q-sequence, a meta-Fibonacci recursion that lacks a known growth law or closed-form solution. Despite its recursive chaos, the values tightly track a diagonal, hinting at regularity buried in self-reference.

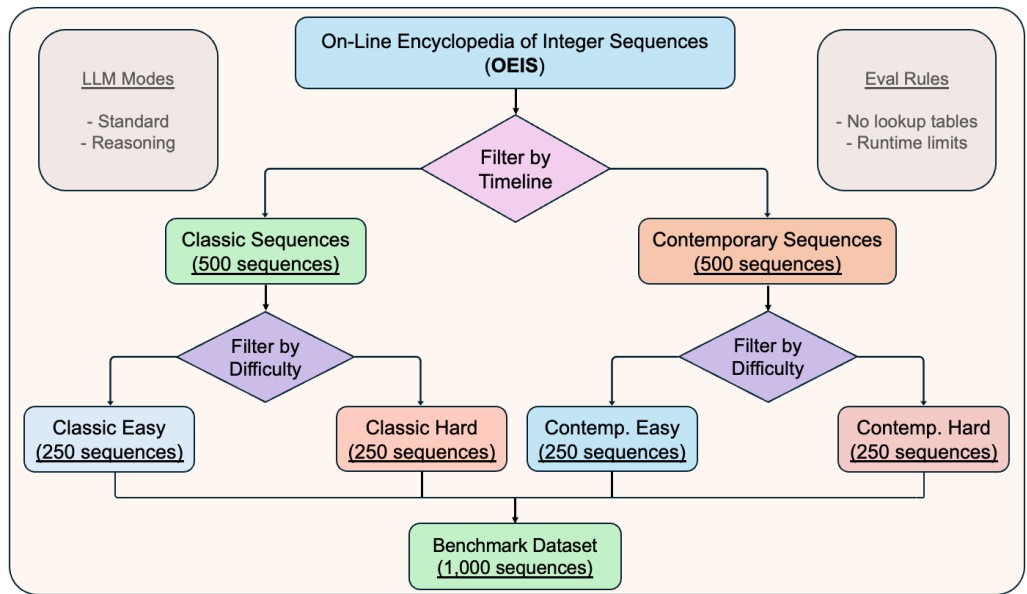

Figure 2: **Workflow for curating the OEIS-based benchmark dataset**. Starting from the full OEIS collection, we first filter by a July 2024 timeline cutoff into "Classic" (pre-cutoff) and "Contemporary" (post-cutoff) sequences. Each branch is then split by the OEIS "easy"/"hard" tags into four subsets: Classic Easy, Classic Hard, Contemporary Easy, and Contemporary Hard, each containing 250 sequences. Finally, these are recombined into the 1,000-sequence benchmark set.

## 3 Benchmark Design

In designing the benchmark, our goal was to create a robust and rigorous evaluation framework that challenges frontier LLMs. The benchmark is centered around writing code that computes elements of integer sequences, using sequences sourced from the OEIS [12]. The design incorporates various levels of difficulty and enforces performance constraints to measure both the accuracy and efficiency of the model-generated code.

### 3.1 Dataset Selection

The dataset for the benchmark is derived from OEIS, an extensive database of integer sequences contributed by a community mathematicians around the world. We selected latest 250 easy and 250 hard sequences based on OEIS labels – around 30 new sequences are added to OEIS every day. The set of sequences is defined as $\mathcal{S} = \mathcal{S}_{easy} \cup \mathcal{S}_{hard}$, where $\mathcal{S}_{easy}$ are 250 recent sequences labeled as easy, and $\mathcal{S}_{hard}$ are 250 recent sequences labeled as hard in OEIS. We also source an additional 250 easy and 250 hard sequences that are the oldest such sequences in OEIS and call these the classic sequences. These classic sequences are included because many of them are of significant mathematical interest (e.g., the first sequence is a number of groups of order $n$, which is fundamental to abstract algebra). The scores for the classic sequences are reported in the appendices. Our discussion in the main text focuses on the contemporary sequences to eliminate the potential for contamination with the models' training data (though scores indicate the models have not been trained to perform well even on the classic sequences).

This selection provides a broad spectrum of sequence generation problems, ranging from basic arithmetic operations to complex mathematical computations. The dataset and the code is available at `https://github.com/ceodspspectrum/oeis-sequence-benchmark`.

### 3.2 Problem Definition

For each sequence $s \in \mathcal{S}$, an LLM $M$ is tasked with generating Python code $C_s$ that computes the first $N$ terms of the sequence $s$, where $N$ is a fixed positive integer (e.g., $N = 10$). Each integer

sequence is a function: $s: \{ i_0 + j \}_{j=0}^{\infty} \to \mathbb{Z}$, where $i_0$ is an offset indicating where the sequence starts. The code $C_s$ should define a function $f_s : \{ i_0 + j \}_{j=0}^{\infty} \to \mathbb{Z}$ such that $f_s(n) = s(n)$ for all $n \geq i_0$. For each sequence, the prompt includes only the OEIS Name and Comments fields; sequence values/formulas are withheld for testing.

The following constraints are imposed on the generated code: (1) the code $C_s$ must not contain a lookup table of the sequence terms, (2) the execution time $t_s$ of $C_s$ must satisfy $t_s \leq T$ where $T$ is a predefined time limit, and (3) the code must be valid Python code executable in a standard environment without external library dependencies. We evaluate the models using $T \in \{0.5, 4\}$ seconds, but these thresholds may need to increase as the models begin to perform better on the benchmark, especially for the hard sequences.

### 3.3 Evaluation Metrics

To provide a comprehensive evaluation of the models, we measure their performance using three factors: accuracy, efficiency, and avoiding lookup tables.

For each sequence $s$, we define the accuracy $A_s(n)$ as:

$$A_s(n) = \{ 0 \ f_s(n) \neq s(n) 0 t_s > T 0 \text{cheating} 1 \text{otherwise} \tag{1}$$

We report the average accuracy over all sequence values in $\mathcal{S}_{easy}$ and $\mathcal{S}_{hard}$. We also report the percentage of sequences where the models correctly compute all sequence values in our test suite for that sequence.

### 3.4 Cheating Detection Mechanism

Another core aspect of the benchmark is ensuring that models produce algorithms rather than lookup tables of sequence values. To enforce this, we use LLM's structured output capabilities (with temperature 0 to maximize reproducibility) to check the code output by the model and flag cases where lookup tables are employed. Any model that is found to be cheating by using a lookup table receives a score of zero for that sequence, regardless of the accuracy of the output. This cheating detection mechanism's effectiveness was validated by comparing it with a human evaluation (one of the authors, who was not provided with the GPT-4o cheating evaluations beforehand). An initial attempt to use GPT-4o in a zero-shot setting achieved 86% accuracy with human evaluators. This was improved by providing GPT-4o with six sequences and their human cheating evaluations to inform its judgment. This increased accuracy to 95% on a fresh set of human evaluations.

## 4 Experiments and Results

We evaluate 21 state-of-the-art LLMs on our integer sequence benchmark using their default settings (temperature, etc.). Table 1 summarizes the models' performance on the contemporary easy ($\mathcal{S}_{easy}$) and hard ($\mathcal{S}_{hard}$) sequence sets and Figure 3 visualizes the performance in detail of the top performing reasoning and non-reasoning models. There are only small differences when the 0.5s and 4s time limits are used, so we focus our discussion on the 4s case. Overall, the o3 model performed best with the highest fraction of perfect scores on sequences for both the easy and hard sequences. Notably, o3-mini had the highest average score (though fewer sequences where it got a perfect score) than regular o3 and o4-mini on the hard sequences. The latest reasoning models from OpenAI (o3, o3-mini,o4-mini) utilize reasoning processes to score above 70% accuracy for easy sequences on average scores and performed better than OpenAI's non-reasoning family of models. Additionally, reasoning models benefit more when they are allowed extra time (4 seconds compared to 0.5 seconds) to execute the code. The latest Gemini (2.5 flash and pro) models performed well compared to older, non-reasoning Gemini models (1.5-flash, 1.5-pro and 2.0-flash).

All models used lookup tables more frequently on the hard sequences than the easy sequences, reminiscent of the adage that "desperate times call for desperate measures." It is also noteworthy that the models with the lowest occurrences of cheating are not the strongest models and there are regressions in cheating from models in the same series. For example, o3 cheated more than o1 on the hard sequences and o4-mini cheated more often than o3-mini on both the hard and easy sequences.

The scores on the classic sequences are reported in Table 2.

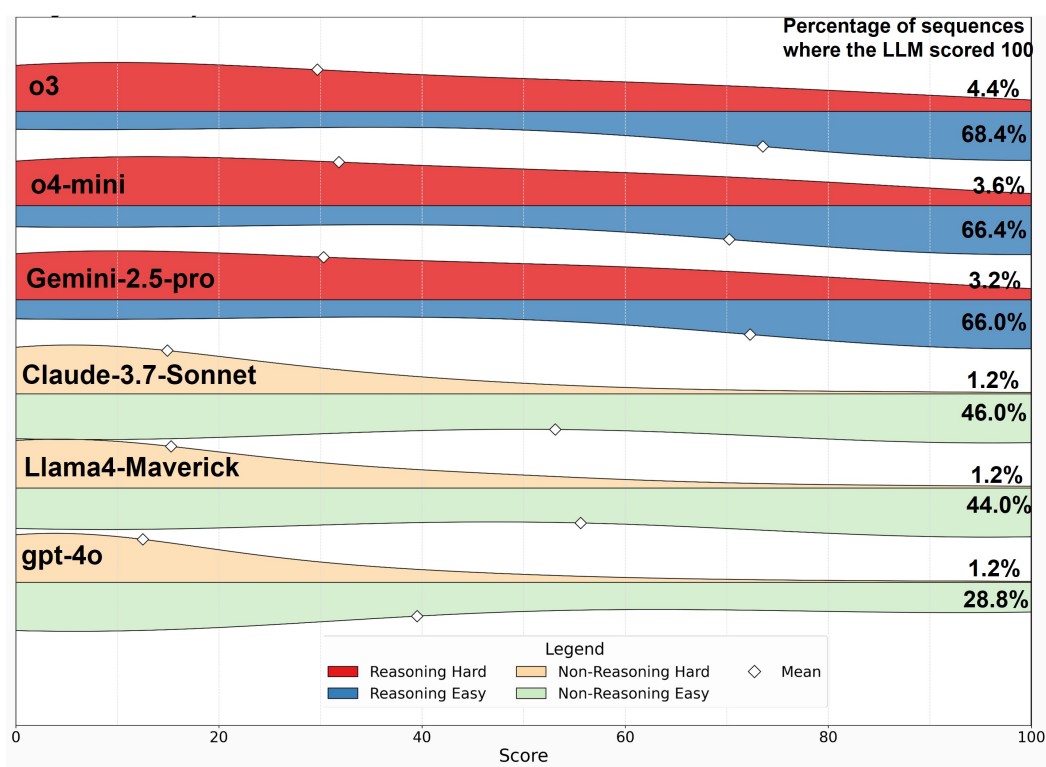

Figure 3: **Distribution of scores for the top three reasoning and non-reasoning models.** Shown are score distributions for the hard sequences (red for reasoning models, yellow for non-reasoning models) and easy sequences (blue for reasoning models, green for non-reasoning models). The percentage of sequences for which each model achieves a perfect score is shown on the right. All models show distributions skewed toward low scores on the hard sequences, while non-reasoning models have near-uniform scores on the easy set and reasoning models are strongly skewed toward high scores.

## 4.1 Case Study of Reasoning vs. Non-Reasoning on sequence A380521

We compare coding solutions between a frontier reasoning model (o3) versus a non-reasoning LLM (LLaMA-405B). We use sequence A380521 ("Primes p such that between p and the next prime there exist 2 distinct integers which are a square and a cube, respectively") for this case study because all frontier reasoning models achieved perfect scores while most non-reasoning models scored 0. We observe that many non-reasoning models produced functional code that correctly calculates the sequence, such as the code for LLaMA-405B (see code 2). However, a key difference emerged in their approach to efficiency. The o3 model demonstrated a deeper algorithmic understanding by implementing memoization (see 1). The o3 model stores previously verified prime numbers to accelerate its prime number checks of future candidates. This reuse of computation significantly reduced redundant work, enabling the solution to execute within the imposed time constraints.

In contrast, the LLaMA-405B model generated a more naive solution with no memoization. The solution of LLaMA-405B led to excessive computation and timeouts. This case exemplifies a broader pattern observed across multiple tasks: reasoning models like o3 typically applied more advanced strategies such as memoization, whereas non-reasoning models often failed to infer these improvements even when producing accurate and valid code. Figure 4 shows how different models tend to utilize memoization and other techniques.

Table 1: **Evaluation of frontier models on the contemporary sequence data split.** Shown are the average accuracy scores, the fraction of sequences for which each model achieves a perfect score, and the fraction flagged for cheating via lookup tables. Reasoning-focused models (e.g., o1, o3, o3-mini, o4-mini, Gemini 2.5-flash/pro) clearly outperform even strong non-reasoning models (e.g., Claude 3.7 Sonnet), with the largest performance gap appearing on the hard sequence split.

| Model | Timeout | SequenceEasy Avg. Score | % Perfect | % Cheating | SequenceHard Avg. Score | % Perfect | % Cheating |
|---|---|---|---|---|---|---|---|
| 2*gpt-3.5-turbo 1106 | 0.5 | 20.0 | 14.0% | 4.0% | 6.6 | 0.0% | 14.8% |
| | 4 | 20.5 | 14.0% | 4.0% | 7.1 | 0.0% | 14.8% |
| 2*gpt-4o | 0.5 | 39.0 | 28.4% | 8.0% | 10.9 | 0.8% | 17.6% |
| | 4 | 39.5 | 28.8% | 8.0% | 12.5 | 1.2% | 17.6% |
| 2*gpt-4o-mini | 0.5 | 34.6 | 27.2% | 6.4% | 11.1 | 0.8% | 18.4% |
| | 4 | 34.6 | 27.2% | 6.4% | 11.6 | 0.8% | 18.4% |
| 2*o1-preview | 0.5 | 55.5 | 47.2% | 5.6% | 19.0 | 2.0% | 18.0% |
| | 4 | 55.8 | 47.2% | 5.6% | 21.5 | 2.8% | 18.0% |
| 2*o1-mini | 0.5 | 57.1 | 48.4% | 2.4% | 19.4 | 1.6% | 12.0% |
| | 4 | 58.1 | 49.2% | 2.4% | 20.9 | 2.0% | 12.0% |
| 2*o1 | 0.5 | 55.5 | 50.8% | 2.8% | 17.7 | 1.6% | 9.2% |
| | 4 | 57.2 | 52.8% | 2.8% | 21.4 | 2.8% | 9.2% |
| 2*o3 | 0.5 | **73.5** | **68.4%** | 2.4% | 26.2 | **3.6%** | 12.0% |
| | 4 | **73.6** | **68.4%** | 2.4% | 29.7 | **4.4%** | 12.0% |
| 2*o3-mini | 0.5 | 70.4 | 64.4% | 2.4% | **29.1** | 2.0% | 8.4% |
| | 4 | 70.5 | 64.4% | 2.4% | **32.0** | 2.0% | 8.4% |
| 2*o4-mini | 0.5 | 70.1 | 66.4% | 5.2% | 28.7 | 3.2% | 14.0% |
| | 4 | 70.3 | 66.4% | 5.2% | 31.8 | 3.6% | 14.0% |
| 2*claude-3.5-sonnet-20241022 | 0.5 | 49.2 | 38.8% | 4.0% | 14.0 | 0.4% | 22.4% |
| | 4 | 49.4 | 38.8% | 4.0% | 14.8 | 0.4% | 22.4% |
| 2*claude-3.7-sonnet-20250219 | 0.5 | 55.5 | 46.0% | 2.8% | 13.7 | 1.2% | 37.6% |
| | 4 | 55.6 | 46.0% | 2.8% | 15.3 | 1.2% | 37.6% |
| 2*llama-405b | 0.5 | 31.8 | 23.2% | 6.8% | 11.4 | 0.4% | 11.6% |
| | 4 | 31.9 | 23.2% | 6.8% | 12.5 | 0.4% | 11.6% |
| 2*llama-70b | 0.5 | 25.7 | 16.4% | 4.8% | 9.9 | 0.4% | 11.6% |
| | 4 | 25.8 | 16.4% | 4.8% | 10.3 | 0.4% | 11.6% |
| 2*llama4-Scout | 0.5 | 37.7 | 28.4% | 7.6% | 12.4 | 0.8% | 23.6% |
| | 4 | 37.7 | 28.4% | 7.6% | 13.2 | 0.8% | 23.6% |
| 2*llama4-Maverick | 0.5 | 53.0 | 44.0% | 9.2% | 13.8 | 1.2% | 20.8% |
| | 4 | 53.1 | 44.0% | 9.2% | 14.9 | 1.2% | 20.8% |
| 2*llama3.3-70b | 0.5 | 32.9 | 24.4% | 4.4% | 10.5 | 0.4% | **7.6%** |
| | 4 | 33.0 | 24.4% | 4.4% | 11.8 | 0.4% | **7.6%** |
| 2*gemini-1.5-flash | 0.5 | 30.3 | 22.8% | 26.0% | 6.4 | 0.8% | 45.2% |
| | 4 | 30.3 | 22.8% | 26.0% | 6.9 | 0.8% | 45.2% |
| 2*gemini-1.5-pro | 0.5 | 32.2 | 23.2% | 16.8% | 6.0 | 0.4% | 66.7% |
| | 4 | 32.3 | 23.2% | 16.8% | 6.4 | 0.4% | 66.7% |
| 2*gemini-2.0-flash | 0.5 | 38.4 | 30.0% | 22.4% | 8.7 | 0.4% | 50.8% |
| | 4 | 38.4 | 30.0% | 22.4% | 9.1 | 0.4% | 50.8% |
| 2*gemini-2.5-flash-preview | 0.5 | 68.7 | 62.4% | **2.0%** | 18.1 | 0.8% | 17.6% |
| | 4 | 69.5 | 62.8% | **2.0%** | 19.6 | 0.8% | 17.6% |
| 2*gemini-2.5-pro-preview | 0.5 | 72.0 | 66.0% | 3.2% | 28.1 | 2.8% | 22.0% |
| | 4 | 72.3 | 66.0% | 3.2% | 30.3 | 3.2% | 22.0% |

## 5   Discussion

The superior performance of reasoning models highlights the effectiveness of specialization in LLMs for mathematical reasoning and coding tasks. The reasoning model's higher accuracy and lower cheating rates demonstrate that models optimized for STEM reasoning can significantly outperform general-purpose models on algorithmic tasks. The low average scores on $\mathcal{S}_{hard}$ across all models

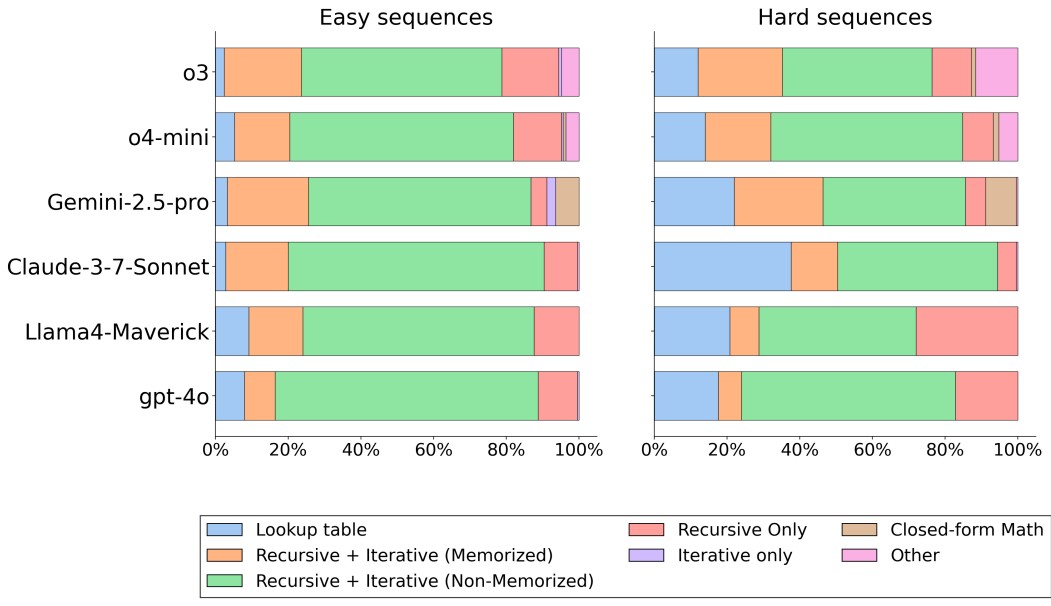

Figure 4: **Classification of error modes for top reasoning and non-reasoning models.** Shown are failure types for three top-performing reasoning and non-reasoning models on both the hard and easy sequence sets. Lookup-table use and memorization occur much more frequently on the hard sequences than on the easy ones.

indicate that current LLMs struggle with generating complex algorithms, emphasizing the need for further advancements in this area. The higher cheating rates in the hard sequences suggest that models may default to memorization when faced with difficult tasks.

Using integer sequence generation from OEIS has proven to be an effective benchmark for evaluating computational reasoning in LLMs. The richness of the OEIS dataset, with its diverse range of sequences, provides a challenge for models across varying levels of difficulty. The sequences test both basic and advanced mathematical concepts, making them ideal for evaluating LLMs' mathematical reasoning and code writing.

Implementing cheating detection mechanisms was instrumental in developing an effective benchmark, given the frequent occurrence of lookup tables in $\mathcal{S}_{hard}$, even for strong models and despite prompting not to use a lookup table. By identifying when models used lookup tables, we ensured that the benchmark tested their ability to generate solutions algorithmically. This mechanism plays a key role in maintaining the integrity of the evaluation.

There are several avenues for future research. Integrating tool use, such as web access combined with retrieval-augmented generation (RAG), could enable models to access additional resources during problem-solving. This could also create problems if, e.g., the models are able to find implementations compatible with our restrictions (e.g., vanilla python that does not depend on advanced mathematical libraries). Allowing LLMs to retrieve and utilize external information, like the extensive references and comments available in OEIS entries, may improve their ability to generate algorithms for complex sequences. In this study, such reference information was not provided to the models. Future variations of this benchmark could incorporate these resources to assess models' abilities to leverage external knowledge effectively. Since the OEIS is continuously updated by a large community, this benchmark can be updated on, say, an annual or semi-annual basis to evaluate progress of generative models on hard math and coding problems while avoiding contamination issues.

# 6 Conclusion

We introduced a rigorous benchmark to evaluate large language models on generating code for integer sequences from the OEIS, focusing on mathematical reasoning and computational efficiency. Our

evaluation demonstrated that reasoning models outperform general-purpose models in tasks requiring mathematical insight and algorithmic coding skills. Specifically, the reasoning models achieved higher accuracy and more perfect scores on both the easy ($\mathcal{S}_{easy}$) and hard ($\mathcal{S}_{hard}$) sequence sets. Despite these strengths, all models showed low performance on the hard sequences, underscoring the challenges LLMs face in generating complex algorithms within practical time constraints. The frequent reliance on memorization strategies, like using lookup tables, despite prompting to avoid it highlights the need for developing models capable of genuine algorithmic reasoning. Our benchmark effectively assessed the computational reasoning abilities of LLMs, with the OEIS dataset providing a robust and diverse evaluation framework. The implemented cheating detection mechanism was crucial in ensuring adherence to algorithmic constraints and maintaining the integrity of the assessment. Importantly, this benchmark can be routinely updated with new sequences added to the OEIS so that the benchmark can always remain ahead of the training data for the models that will be evaluated on it.

# 7 Limitations

Our benchmark possesses several limitations that warrant consideration. First, relying exclusively on the OEIS as the source of integer sequences may introduce biases due to the specific types and distributions of sequences included, as well as their subjective labeling as "easy" or "hard." Another potential issue is that some OEIS sequences have associated code snippets that are publicly available. While this could assist LLMs in generating the correct sequence, many of the selected problems are difficult enough to require novel mathematical insights or optimized coding techniques. The codes that are available in the OEIS database are most often in languages like Mathematica, Maple, or Magma and tend to rely on advanced functionality of these pieces of software that can turn complex sequences into a few lines of code. To mitigate this, we require the models to generate code using the Python standard library, which does not have equivalent functionality. In practice, current models face considerable challenges in computing the sequences efficiently, especially for the $\mathcal{S}_{hard}$ problems. The OEIS is also actively maintained with 30-60 new sequences being added on a daily basis [10]. So, the benchmark could be continuously updated to mitigate the effect of OEIS sequence information being included in the training of LLMs.

Second, although our cheating detection mechanism effectively identifies the use of lookup tables, it is not infallible. With a $95\%$ agreement rate with human evaluators, some instances of cheating may go undetected or be falsely flagged, potentially impacting the accuracy of the evaluation. Of course, there is fundamentally some subjectivity in the determination of whether or not a code uses a lookup table.

Third, restricting code generation to Python confines the evaluation to a single programming language. This limitation may not fully capture a model's versatility or efficiency in other languages that could be more suitable for certain sequences. Models might perform differently if allowed to utilize languages better aligned with the computational demands of specific tasks. Python has a relatively slow computational speed. At the same time, many sequences have a seemingly high cost to compute some of the sequence values. For example, A000791 [11] is a sequence in $\mathcal{S}_{hard}$ of Ramsey numbers – notoriously difficult to compute – and one of the comments notes that the tenth element in the sequence being 42 was ruled out "with a massive computer search." This combination of a slow language and expensive computations suggests that a different program language might provide the models a better chance of success.

Fourth, the imposed time constraints, while essential for assessing efficiency, may disadvantage models that implement correct but computationally intensive algorithms, especially for sequences inherently requiring significant resources. This could unfairly penalize models due to factors beyond their control, such as hardware limitations or the intrinsic complexity of the problem.

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

## Appendix A: Classic Sequence Evaluation

The scores on the classic sequences are shown in Table 2.

## Appendix B: Code Examples

Listing 1: o3 Reasoning Model Solution for sequence A380521

```python
import sys
import math

def icbrt(n: int) -> int:
    """
    Integer cube root: floor of real cube root of n
    """
    lo, hi = 0, int(n ** (1/3)) + 2
    while lo < hi:
```

Table 2: The scores on the classic sequences are shown.

| Model | Timeout | SequenceEasy | | | SequenceHard | | |
|---|---|---|---|---|---|---|---|
| | | Avg. Score | % Perfect | % Cheating | Avg. Score | % Perfect | % Cheating |
| 2*o1 | 0.5 | 57.8 | 51.4% | 2.4% | 16.0 | 1.2% | 11.6% |
| | 4 | 57.8 | 51.4% | 2.4% | 17.4 | 1.2% | 11.6% |
| 2*o3 | 0.5 | 81.7 | 74.6% | 1.2% | 24.3 | 3.3% | 16.4% |
| | 4 | 81.8 | 74.6% | 1.2% | 25.4 | 3.3% | 16.4% |
| 2*o3-mini | 0.5 | 77.5 | 67.2% | 0.4% | 23.0 | 2.0% | 20.8% |
| | 4 | 77.8 | 67.6% | 0.4% | 24.2 | 2.4% | 20.8% |
| 2*o4-mini | 0.5 | 80.0 | 73.6% | 2.4% | 24.4 | 2.8% | 25.6% |
| | 4 | 80.1 | 73.6% | 2.4% | 25.7 | 2.8% | 25.6% |
| 2*claude-3-7-sonnet-20250219 | 0.5 | 60.7 | 53.6% | 2.4% | 9.1 | 1.2% | 55.0% |
| | 4 | 60.7 | 53.6% | 2.4% | 9.6 | 1.2% | 55.0% |
| 2*llama4-Scout | 0.5 | 50.5 | 42.4% | 4.8% | 9.0 | 0.8% | 40.8% |
| | 4 | 50.6 | 42.4% | 4.8% | 9.3 | 0.8% | 40.8% |
| 2*llama4-Maverick | 0.5 | 59.6 | 48.4% | 3.2% | 10.7 | 0.8% | 41.9% |
| | 4 | 59.7 | 49.2% | 3.2% | 11.3 | 0.8% | 41.9% |
| 2*llama3.3-70b | 0.5 | 52.1 | 45.6% | 4.4% | 9.2 | 0.4% | 21.6% |
| | 4 | 52.1 | 46.0% | 4.4% | 9.5 | 0.4% | 21.6% |
| 2*gemini-2.0-flash | 0.5 | 56.2 | 50.0% | 14.4% | 6.5 | 0.4% | 61.6% |
| | 4 | 56.3 | 50.0% | 14.4% | 6.7 | 0.4% | 61.6% |
| 2*gemini-2.5-flash-preview-04-17 | 0.5 | 72.4 | 67.2% | 0.0% | 14.3 | 1.6% | 26.4% |
| | 4 | 72.4 | 67.2% | 0.0% | 15.0 | 1.6% | 26.4% |
| 2*gemini-2.5-pro-preview-03-25 | 0.5 | 77.4 | 70.4% | 3.2% | 15.5 | 1.6% | 47.2% |
| | 4 | 77.4 | 70.4% | 3.2% | 16.3 | 1.6% | 47.2% |
| 2*gpt-3.5-turbo-1106 | 0.5 | 35.8 | 30.0% | 0.0% | 6.5 | 0.4% | 0.0% |
| | 4 | 35.9 | 30.0% | 0.0% | 6.6 | 0.4% | 0.0% |
| 2*gpt-4o | 0.5 | 54.5 | 48.8% | 4.0% | 8.7 | 0.4% | 37.2% |
| | 4 | 54.6 | 48.8% | 4.0% | 8.9 | 0.4% | 37.2% |
| 2*gpt-4o-mini | 0.5 | 51.6 | 45.6% | 6.0% | 8.5 | 0.8% | 32.0% |
| | 4 | 52.0 | 45.6% | 6.0% | 8.7 | 0.8% | 32.0% |
| 2*o1-preview | 0.5 | 63.2 | 55.2% | 2.8% | 18.0 | 1.2% | 23.2% |
| | 4 | 63.2 | 54.8% | 2.8% | 18.7 | 1.2% | 23.2% |
| 2*o1-mini | 0.5 | 65.3 | 58.4% | 2.0% | 17.3 | 1.6% | 15.2% |
| | 4 | 65.4 | 58.4% | 2.0% | 18.1 | 2.0% | 15.2% |
| 2*claude-3-5-sonnet-20241022 | 0.5 | 56.9 | 50.8% | 2.0% | 10.8 | 0.8% | 42.0% |
| | 4 | 57.0 | 51.2% | 2.0% | 11.1 | 0.8% | 42.0% |
| 2*llama-405b | 0.5 | 47.4 | 39.6% | 4.4% | 9.9 | 0.8% | 32.0% |
| | 4 | 47.6 | 40.0% | 4.4% | 10.1 | 0.8% | 32.0% |
| 2*llama3.1-70b | 0.5 | 44.7 | 36.4% | 2.4% | 8.6 | 0.4% | 22.0% |
| | 4 | 44.7 | 36.8% | 2.4% | 8.7 | 0.4% | 22.0% |
| 2*gemini-1.5-flash | 0.5 | 45.9 | 37.6% | 4.0% | 6.5 | 0.8% | 36.0% |
| | 4 | 46.1 | 38.0% | 4.0% | 6.6 | 0.8% | 36.0% |
| 2*gemini-1.5-pro | 0.5 | 49.4 | 45.2% | 20.8% | 4.7 | 0.4% | 73.2% |
| | 4 | 49.4 | 45.2% | 20.8% | 4.9 | 0.4% | 73.2% |

```
        mid = (lo + hi + 1) // 2
        if mid * mid * mid <= n:
            lo = mid
        else:
            hi = mid - 1
    return lo

def has_square_and_cube(p: int, q: int) -> bool:
    """
    Returns True iff the open interval (p, q) contains
```

```python
        two distinct integers , one a perfect square and the other a
            perfect cube .
        """
    if q - p < 3:
        return False

    first_sq_root = math.isqrt(p) + 1
    last_sq_root  = math.isqrt(q - 1)
    if first_sq_root > last_sq_root:
        return False

    first_cu_root = icbrt(p) + 1
    last_cu_root  = icbrt(q - 1)
    if first_cu_root > last_cu_root:
        return False

    if (first_sq_root == last_sq_root and
            first_cu_root == last_cu_root and
            first_sq_root ** 2 == first_cu_root ** 3):
        return False

    return True

def generate_sequence_element(n: int) -> int:
    """
    Generates the n-th prime p for which the open interval (p,
        next_prime)
    contains a square and a cube (at least one of each , and they are
        distinct).
    """
    if n <= 0:
        raise ValueError("n must be a positive integer")

    primes = [2]
    seq_count = 0
    prev_prime = 2
    candidate = 3

    while True:
        root = int(math.isqrt(candidate))
        is_p = True
        for pr in primes:
            if pr > root:
                break
            if candidate % pr == 0:
                is_p = False
                break
        if is_p:
            primes.append(candidate)
            if has_square_and_cube(prev_prime , candidate):
                seq_count += 1
                if seq_count == n:
                    return prev_prime
            prev_prime = candidate
        candidate += 2

def main():
    try:
        n = int(sys.stdin.readline().strip())
    except Exception:
        return
    result = generate_sequence_element(n)
    print(str(result))
```

```
if __name__ == "__main__":
    main()
```

Listing 2: llama405B Model Solution for sequence A380521

```python
def is_prime(n):
    """Check if n is a prime number."""
    if n < 2:
        return False
    for i in range(2, int(n ** 0.5) + 1):
        if n % i == 0:
            return False
    return True

def is_square(n):
    """Check if n is a perfect square."""
    return int(n ** 0.5) ** 2 == n

def is_cube(n):
    """Check if n is a perfect cube."""
    return round(n ** (1. / 3)) ** 3 == n

def next_prime(n):
    """Return the next prime number greater than n."""
    n += 1
    while not is_prime(n):
        n += 1
    return n

def check(n):
    """
    Check if the open interval (n, next_prime(n)) contains
    a square followed by a distinct cube.
    """
    next_n = next_prime(n)
    for i in range(n + 1, next_n):
        if is_square(i):
            for j in range(i + 1, next_n):
                if is_cube(j):
                    return True
    return False

# Read input
n = int(input())
count = 0
num = 2

# Find the nth qualifying prime
while True:
    if is_prime(num) and check(num):
        count += 1
        if count == n:
            print(num)
            break
    num += 1
```

## Appendix C: Prompts

Here, we provide the prompts we used for code generation and lookup table detection using Python's string interpolation syntax. The actual prompts would contain the interpolated information.

**Code Generation Prompt**

f"""Write a python code that takes a number n as input from stdin, outputs a single element of a sequence as a string to stdout. Just output the n-th element in the sequence, not the whole sequence or anything other than the single sequence element. You will be graded by my running your code and comparing the results with a lookup table. If you use a lookup table for your code, you will not only fail the test. Hardcoding values that can be computed without hardcoding is considered to be using a lookup table. Use only the standard python library. No packages will be installed with pip or conda. Output your final code at the end of your response inside triple backticks like:\n'''\n#your code goes here\n'''\nThe last part of your response that is inside triple backticks will be used as your response for the purposes of the test. The code will be stopped after a short period of time, so make it efficient if needed.\n\nHere is some information on the sequence:\n'''\nName: {info["name"]}\nComments: {info["comments"]}\n'''"""

## NeurIPS Paper Checklist

