# OpenReview forum: "Benchmarking Large Language Models with Integer Sequence Generation Tasks"
_NeurIPS.cc/2025/Datasets_and_Benchmarks_Track — NeurIPS 2025 Datasets and Benchmarks Track poster_

### Official Review · Reviewer_K1Ma · 2025-06-21

**Rating:** 4
**Confidence:** 2

**Summary:**

This paper introduces a new benchmark for evaluating the mathematical reasoning and algorithmic code generation capabilities of LLM.  The benchmark is constructed using 1,000 integer sequence generation tasks sourced from the Online Encyclopedia of Integer Sequences (OEIS).  The sequences are categorized into "easy" and "hard" based on OEIS tags, and are also divided into "classic" (older) and "contemporary" (recent) sets to mitigate data contamination from model training sets.
The core task for an LLM is to generate efficient and accurate Python code to compute the terms of a given sequence without resorting to lookup tables.  A key contribution is an automated "cheating detection" mechanism, validated against human evaluation, to flag and penalize models that use memorized sequence values instead of generating a true algorithm.  The study benchmarks a wide array of 21 leading LLMs, including reasoning-specialized models (like OpenAI's o-series and Google's Gemini 2.5-pro) and general-purpose models.  The results show that reasoning-focused models significantly outperform their non-reasoning counterparts, especially on easy tasks.  However, all models struggle significantly with the "hard" sequences, indicating that complex algorithmic reasoning remains a major challenge for the current state-of-the-art.

**Dataset Code Accessibility:**

Yes

**Ethical Considerations:**

No, there are no or only very minor ethics concerns

**Final Justification:**

As the authors address most of my concerns in the rebuttal, I decide to lean to acceptance.

**Limitations Weaknesses:**

1. About Cheating Detection: While innovative, the cheating detection mechanism is acknowledged to be imperfect, with a 5% disagreement rate with human evaluators.  This could lead to a small but non-zero number of misclassified submissions, potentially affecting the accuracy of the overall results. The paper concedes that there is "fundamentally some subjectivity" in determining whether a piece of code is using a lookup table. This ambiguity makes creating a perfect, automated judge difficult.

2. Potential Data Contamination: The authors have taken commendable steps to mitigate data contamination by using recently added OEIS sequences.  However, they also acknowledge that some OEIS sequences have publicly available code snippets in other languages, which could have been part of the models' training data.  While the task requires translation to standard Python, the underlying logic might still be learned from these existing solutions, a possibility that is difficult to completely rule out.

**Strengths Contributions:**

The use of integer sequence generation from the OEIS is an excellent and creative way to test the algorithmic reasoning of LLMs.  It moves beyond standard math or coding benchmarks by requiring a blend of pattern recognition, mathematical insight, and the ability to translate that insight into efficient code.  The dataset's design, splitting sequences by difficulty ("easy"/"hard") and time ("classic"/"contemporary"), is thoughtful and adds to the benchmark's robustness.

The results clearly delineate the performance gap between reasoning and non-reasoning models and highlight the persistent difficulty of complex algorithmic tasks.  Furthermore, because the OEIS is continuously updated, the authors have proposed a benchmark that can evolve and stay ahead of model training data, ensuring its long-term utility.

---

> ### Author Rebuttal · Authors · 2025-07-30
>
> Thank you for your feedback and supportive comments. We appreciate your statement that our approach "is an excellent and creative way to test the algorithmic reasoning of LLMs." We also appreciate the points your raise in the limitations. We address each of your concerns below.
>
> **Concern: Imperfection of cheating detection**: While no automatic judge is perfect, a 95% agreement results in a very small variance in the scores (see mathematical derivation below). We don't think that the small noise associated with the judge is preventing us from detecting a step change in model capabilities, which is needed to obtain good results on the hard sequences.
>
> **Concern: Data Contamination**: Regarding data contamination: As you mention we have taken "commendable steps to mitigate data contamination." Particularly, the use of "contemporary" sequences, which are continually added to OEIS (~30 per day, allowing us to continually update the benchmark as model training cutoff dates move forward),  effectively eliminates data contamination risks. In the context of the "classic" sequences, the point you raise is a potential issue. We want to emphasize that for many "classic" sequences, the codes that are included provide essentially no information about how to implement them in python. For the first sequence in our benchmark (A000001: Number of groups of order n.), the codes that are included in OEIS are
> ```
> MAPLE
> GroupTheory:-NumGroups(n); # with(GroupTheory); loads this command - N. J. A. Sloane, Dec 28 2017
> MATHEMATICA
> FiniteGroupCount[Range[100]] (* Harvey P. Dale, Jan 29 2013 *)
> a[ n_] := If[ n < 1, 0, FiniteGroupCount @ n]; (* Michael Somos, May 28 2014 *)
> PROG
> (Magma) D:=SmallGroupDatabase(); [ NumberOfSmallGroups(D, n) : n in [1..1000] ]; // John Cannon, Dec 23 2006
> (GAP) A000001 := Concatenation([0], List([1..500], n -> NumberSmallGroups(n))); # Muniru A Asiru, Oct 15 2017
> ```
> These are basically restating the definition. Picking a random example (A002860: Number of Latin squares of order n; or labeled quasigroups.) has the code
> ```
> MATHEMATICA
> Table[Length[ResourceFunction["FindProperColorings"][GraphProduct[CompleteGraph[n], CompleteGraph[n], "Cartesian"], n]], {n, 5}]
> ```
> which would not be very helpful in constructing a python implementation that only uses the standard library. So, the contamination issues are mitigated by the fact that many classic sequences are of significant mathematical importance that they have extremely concise implementations languages like Maple, Mathematica, Magma, etc. that are often used with OEIS.
>
> ## Mathematical derivations
> Let $C_i$ be 1 if the code passes tests and 0 otherwise. Let $J_i$ be 1
> if the judge determines no look-up table is used and 0 otherwise. Let
> $H_i$ be 1 if a human judge determines no look-up table is used and 0
> otherwise. The score in our tables is given by
> $$S_{J} = \frac{1}{N}\sum_{i=1}^N{C_i J_i}$$ where $N=250$. Using a
> human judge, the score would be
> $$S_H = \frac{1}{N}\sum_{i=1}^N{C_i H_i}$$ where $N=250$. The difference
> between the two scores is
> $$D = S_J - S_H = \frac{1}{N}\sum_{i=1}^N{C_i (J_i - H_i)}$$
>
> Our data is consistent with the hypotheses that
> $Pr(J_i=1 \cap H_i=0) = Pr(J_i=0 \cap H_i=1)$ and
> $E[C_i|J_i=1 \cap H_i=0]=E[C_i|J_i=0 \cap H_i=1]$. Using these results, we get $$\begin{aligned}
>     E[D] &=& E\left[\frac{1}{N}\sum_{i=1}^N{C_i (J_i - H_i)}\right] \\
>          &=& E[0|J_i=H_i] Pr(J_i=H_i) + E[C_i|J_i=1 \cap H_i=0] Pr(J_i=1\cap H_i=0) - E[C_i|J_i=0\cap H_i=1] Pr(J_i=0\cap H_i=1)\\
>          &=& 0
> \end{aligned}$$ This indicates that our estimator is unbiased. Next, we
> examine the variance of the estimator. Let
> $1-Pr(J_i=H_i)=\epsilon\approx0.05$ based on our data. $$\begin{aligned}
>     \mathrm{Var}[D] &=& \mathrm{Var}\left[\frac{1}{N}\sum_{i=1}^N{C_i (J_i - H_i)}\right] \\
>     &=& \frac{1}{N^2}\sum_{i=1}^N{\mathrm{Var}\left[C_i (J_i - H_i)\right]} \\
>     &=& \frac{1}{N^2}\sum_{i=1}^N{\mathrm{E}\left[C_i^2 (J_i - H_i)^2\right]} \\
>     &=& \frac{1}{N} E[C_i^2(J_i-H_i)^2|J_i=H_i]Pr(J_i=H_i) + E[C_i^2(J_i-H_i)^2|J_i \ne H_i]Pr(J_i \ne H_i) \\
>     &=& \frac{\epsilon}{N} E[C_i^2|J_i \ne H_i] \\
>     &=& \frac{\epsilon}{N} Pr[C_i=1|J_i \ne H_i]
> \end{aligned}$$
>
> The variance induced by the imperfect judge is proportional to
> $\epsilon$ divided by the number of samples, which is very small.

---

### Official Review · Reviewer_6xJx · 2025-06-30

**Rating:** 5
**Confidence:** 4

**Summary:**

This work benchmarks the ability of LLMs to generate integer-valued sequences. The dataset is collected from the OEIS. For each task, the model is provided with a natural language description and comments of a sequence and is asked to generate Python code to programmatically produce it. To prevent trivial memorization via lookup tables, a cheat detection mechanism is also proposed.

**Dataset Code Accessibility:**

Yes

**Ethical Comments:**

No, there are no ethics concerns.

**Ethical Considerations:**

No, there are no or only very minor ethics concerns

**Final Justification:**

The author has addresed my concerns on data curation, thus, I would like to increase my score.

**Limitations Weaknesses:**

1. The choice of sequences seems overly simplistic or arbitrary. The current method iterate through the indices and selects the first 500 sequences labeled as "easy" or "hard" without any human validation or curation. This could result in a dataset that is unbalanced or unrepresentative. Some sequences might be trivially easy, while others could be so difficult that even humans lack known algorithms to generate them. A more careful selection process, possibly with human validation would improve the quality and reliability of the benchmark.

2. It would be interesting to explore whether models can revise or self-correct their answers when given feedback. For example, once the model generates an incorrect sequence, could it improve its output if shown the correct sequence and asked to adjust its code? This would evaluate the model’s problem-solving ability, not just its initial generation.

**Strengths Contributions:**

* The paper is well-written and easy to follow.
* The task is well-defined: only descriptions and comments are provided to the model—closed-form formulas or terms of the sequence are not revealed.
* A range of LLMs is evaluated, providing insights into their relative strengths in this kind of symbolic reasoning task.
* Situations where codes generate lookup tables are consider.

---

> ### Author Rebuttal · Authors · 2025-07-30
>
> Thank you for taking the time to review our paper and help us improve it. We appreciate your comments that the paper is well-written, the task is well-defined, etc.  We address each of your concerns below.
>
> **Concern: Curation and representation**: Every OEIS entry is peer-reviewed by editors, so the underlying data is already human-validated. Each sequence undergoes a peer review process before being added to OEIS. We appreciate your concern about balance. In addition to human curating and editing, the balance and representation is better than what is described in your comment. We select the first 250 hard and 250 easy sequences and call these the "classic" sequences. Many of these are important mathematically and they cover a wide range of territory -- lots of diversity there. On top of these, we take 250 easy recent sequences and 250 hard recent sequences. These are often of less mathematical significance, but it allows us to avoid contamination issues. Since ~30 sequences are added daily to OEIS, we can continually update the benchmark to stay ahead of model training cutoff dates. To avoid over-representation in the contemporary sequences, we automatically filtered out clusters of  closely related sequences, which are sometimes submitted together.
>
> **Concern: Going beyond the initial generation**: We agree with this point. We are excited to see how agentic workflows, recent IMO-gold winning models, etc. perform. However, we think this work is outside the scope of our paper. Similar to our comment about prompt engineering to reviewer V8ek, we think that these engineering efforts to boost the performance are interesting avenues for future research (and needed given the poor performance of the models on the hard sequences). Our goal in this paper is to provide the means to evaluate models and any agentic scaffolding on this benchmark.

---

> > ### Comment · Reviewer_6xJx · 2025-08-01
> >
> > Thank the authors for addresing my concerns, I would like to increase my initial score.

---

### Official Review · Reviewer_YcJs · 2025-07-02

**Rating:** 5
**Confidence:** 4

**Summary:**

This paper presents a new benchmark for seeing whether LLMs can write the functions necessary to generate easy and hard integer sequences derived from om the Online Encyclopedia of Integer Sequences. The benchmark includes mostly very new sequences along this easy/hard spectrum, and it provides a clever mechanism to detect cheating and other hacks. The best models get some traction on this benchmark, but it's still very hard. There's also a nice case study of an algorithmically challenging sequence that the best models can handle easily but that other models struggle with even when they write functional but inefficient code.

**Dataset Code Accessibility:**

Yes

**Dataset Code Comments:**

A public link to the dataset is included, and the dataset itself was included in the supplementary materials.

**Ethical Considerations:**

No, there are no or only very minor ethics concerns

**Final Justification:**

I am keeping my initial score. I like the paper. I am not totally in support of the task design, though. On balance, this plus the other input from the authors leaves me back at my initial score.

**Limitations Weaknesses:**

I feel unsure about a quite basic thing: what information is given to the model as input? For the entire time I was reading the main text of the paper, I assumed that they were presented with integer sequences and had to infer what function computed that sequence. However, the prompt in Appendix C seems to indicate that only the "name" and "comments" keys are part of the input, not the integer sequence itself. Is this correct? It does disambiguate sequences that may be ambiguous (Wittgenstein would say every sequence is ambiguous), but then it seems like models often were not given enough information to solve the problem unless they knew the coding scheme. For example, the A374527.json has:

```py
"name": "Odd numbers whose Collatz trajectory is a Sidon sequence.",
"comment": ["For more information see A375006."],
```

To succeed here, one needs to know how to use these keywords, not how to analyze the following sequence for efficient functions to compute it:

```py
"data":"1,21,85,151,227,341,1365,5461,14563,21845,87381,349525,932067,1398101,5592405,22369621,26512143,39768215,59652323,89478485,357913941,1431655765,3817748707,5726623061
```

Please do let me know if I am misunderstanding something.

**Strengths Contributions:**

I like this paper. The benchmark is very creative and seems like it will be very informative. It's also nicely extensible -- new sequences added to the OEIS can be added to the benchmark. I predict that I will use the benchmark myself in my own research.

I think the paper is well-scoped: it introduces the benchmark and provides a large number of initial results. With luck, this will lead to innovations that push things forward for the hard sequences and, in turn, for coding tasks in general.

---

> ### Author Rebuttal · Authors · 2025-07-30
>
> Thank you for your insightful comments. We hope that your prediction about using this benchmark in your research comes true. We address your concern below.
>
> **Concern: What is provided to the model?**: You correctly identified that only
>  the name and the comments are provided to the model, not the numerical sequence itself. We will clarify this in an updated version of the paper. As you rightly noted,this approach requires the model to rely on general knowledge (eg. knowing what a "Collatz trajectory" or a "Sidon sequence" is). However, these models have extremely broad knowledge and this is often the easy part for the model (e.g., we prompted several frontier and several older models and all correctly answered the question "What is a Collatz trajectory?" on the first try). A benchmark based on providing the sequence values could also be very interesting. However, we feel that going by the defition is the better approach. First, the definition is how a person would think of the sequence, so by boosting performance on this task, we can make the models more useful to people. Second, by providing only the definition, we withhold the numerical values that will be used to test the model. This increases the difficulty of things like reward hacking.
>
> We will add this clarification to section 3.2 (Problem Definition).

---

### Official Review · Reviewer_V8eK · 2025-07-02

**Rating:** 5
**Confidence:** 4

**Summary:**

This paper introduces a new benchmark for evaluating large language models (LLMs), focusing on their mathematical reasoning and algorithmic code-generation skills. The benchmark is constructed from 1,000 integer sequences drawn from the On-Line Encyclopedia of Integer Sequences (OEIS). These sequences are divided into “easy” and “hard” subsets, and further classified as “classic” or “contemporary” to mitigate training-data contamination. As this benchmark evolves, the paper argues that it will offer a holistic evaluation of frontier LLMs. The primary task for the models is to generate Python code that computes sequence terms without relying on lookup tables. A key contribution is an automated “cheating-detection” mechanism—validated by human annotators—to ensure that models produce genuine algorithms. The study evaluates 21 state-of-the-art models, finding that specialized reasoning models (such as OpenAI’s o-series and Google’s Gemini 2.5-pro) significantly outperform non-reasoning models, especially on the easy tasks. However, all models struggle with the hard sequences, underscoring persistent challenges in advanced algorithmic reasoning.

**Additional Feedback:**

The evaluation paradigm is my only concern about the paper. The approach, benchmark and the task itself is quite novel.

**Dataset Code Accessibility:**

Partly

**Dataset Code Comments:**

While the authors did release the dataset, evaluation scripts, exact prompts have not been released. Since the authors have not discussed major prompting tweaks or changes I think this is fine but ideally they should release the entire dataset, code and evaluation scripts for reproducibility.

**Ethical Considerations:**

No, there are no or only very minor ethics concerns

**Final Justification:**

Updated rating: Very satisfied with the authors' explanation. Even though I disagree with some of the conclusions this is a worthy contribution to the field.

**Limitations Weaknesses:**

The authors have already noted that the easy/hard categorization can’t be fully trusted, since some problems may appear in the training set or have publicly available solutions. This is worth highlighting since some of these problems would definitely appear in the training dataset of the model and the exact performance of the model will be hard to tease out on the difficulty of the task.

Personally, the biggest weakness is the lack of varied prompting techniques in the evaluation. With so many innovations in prompting and newer search-based approaches, it’s unclear whether the current methodology truly reflects each model’s capabilities. For example, Llama-4 Scout scores an average of 50.5, compared to Llama 3.3 70B’s 52. Would using prompts tailored to Llama-4’s strengths yield better results than those for Llama 3.3? Given the ample public evidence that Llama-4 excels at reasoning (see its Hugging Face benchmark here (https://huggingface.co/meta-llama/Llama-4-Scout-17B-16E) ), could alternative prompting or task formulations alter these conclusions?

**Strengths Contributions:**

The paper has many strengths. It is well-organized, easy to read, and thoughtfully written.

I particularly appreciate its emphasis on evolving benchmarks. As frontier models improve and consume ever-larger amounts of web-crawled data, a continuously updating benchmark will more accurately reflect their true capabilities.

I also found it insightful that FrontierMath—a recently released benchmark—was deliberately designed with problems solvable by humans, whereas this new benchmark includes challenges for which no human solutions are currently known. That distinction makes it an excellent catalyst for the community to advance and rigorously assess LLMs’ mathematical skills.

Finally, the paper highlights how state-of-the-art LLMs can employ diverse algorithmic strategies to tackle these sequences. For example, the code snippets in lines 161–163 not only demonstrate problem-solving ability but also showcase efficient implementations—an ideal use case for spotlighting reasoning models’ strengths.

---

> ### Author Rebuttal · Authors · 2025-07-30
>
> Thank you for your thoughtful review. We appreciate your acknowledgement of our evolving  benchmark and its value in staying ahead of frontier model training data. We address each of your concerns as follow.
>
> **Concern: varied prompting techniques and relative performance of Llama models**: We intentionally used a single canonical prompt to ensure a fair and consistent comparison across all the models. We agree that each model's performance could be improved by engineering a prompt specifically for that model (and many other techniques). Similar to our response to Reviewer 6xJx, we view these types of optimizations (as well as agentic scaffolding, etc.) as being outside the scope of our paper. Our benchmarks explicitly aims to provide a robust foundation upon which future research can build and explore such enhancements, particularly given the significant opportunity for the improvement highlighted by the  poor performance  on the hard sequences.
>
> Regarding to the  linked table: Llama 3.3 70B indeed outperforms Llama-4 Scout in 2 out of the 4 listed benchmarks. Importantly, among these benchmarks, LiveCodeBench is the  most similar to ours, involving coding tasks and explicitly avoiding contamination, just like our contemporary sequences. On LiveCodeBench, Llama 3.3 70B superior performance aligns precisely with our results. Therefore, we believe the  relative performance observed on our benchmark aligns well with prior evaluations and should not raise a concern.
>
> **Concern: Imperfection of easy/hard categorization**: We agree with your point that the easy/hard categorization is heuristic and inherently imperfect. However, we believe this distinction remains highly informative, as evidenced by substantial gap (>60 points) in model performance observed between the easy and hard sequences. To address your concern, we will  explicitly acknowledge the heuristic nature of this categorization in our revision while also pointing out its practical value.

---

### Note · Authors · 2025-08-14

We thank the reviewers for their thoughtful feedback. Overall, we showed that all current models struggle to generate code for the hard tasks in our benchmark, indicating a clear research gap. By providing a publicly-available benchmark for challenging math/coding problems, we enable progress on algorithmic reasoning in AI models. Reviewers liked the benchmark’s novelty, emphasis on algorithmic code generation, and contamination-resistance. One reviewer explicitly noted they expect to use the benchmark in their own research. Three high-confidence initial reviews supported acceptance, and one of these indicated they would increase their score. One additional review was more tentative (score 3; confidence 2). In our rebuttals, we addressed all the issues raised by the reviewers and have not received any pushback on the rebuttals.

## Clarifications & Commitments

* **Task design:** The model input is the OEIS name and comments only (no sequence values). This matches how humans reason from definitions and mitigates reward-hacking. We will revise section 3.2 to clarify that sequence values are not provided to the model to avoid confusion on this issue.
* **Prompting choice & Llama performance:** A single canonical prompt was used to ensure comparability across models. Model-specific prompting and agentic scaffolding were considered out of scope for this benchmark release.  Related to this, we also clarified why our Llama-3.3 70b vs. Llama-4 Scout ordering is consistent with contamination-aware coding benchmarks such as LiveCodeBench.
* **Cheating detection.** The automatic judge agrees with human annotations 95% of the time. In the rebuttal we provided a short derivation showing the scores in our benchmark are unbiased and that the residual variance due to the discrepancy between the automatic judge and a human judge is small at our sample sizes.
* **Curation and contamination.** OEIS entries are reviewed by a human editor. Our dataset combines classic and contemporary sequences and filters clusters of closely related contemporary submissions. Around 30 sequences/day are added to OEIS, enabling rolling updates to our benchmark. We explained why this design mitigates  contamination for the contemporary sequences and the reviewers applauded this aspect of the benchmark.
* **Difficulty labeling.** We will explicitly acknowledge in the paper that the easy/hard categorization is heuristic, while noting its practical value given the observed large performance gap.

---

### Decision · Program_Chairs · 2025-09-18

**Decision:**

Accept (poster)

**Comment:**

This paper introduces a new benchmark for evaluating the mathematical reasoning and algorithmic code generation capabilities of LLM. The benchmark is constructed using 1,000 integer sequence generation tasks sourced from the Online Encyclopedia of Integer Sequences (OEIS). The sequences are categorized into "easy" and "hard" based on OEIS tags, and are also divided into "classic" (older) and "contemporary" (recent) sets to mitigate data contamination from model training sets.
The core task for an LLM is to generate efficient and accurate Python code to compute the terms of a given sequence without resorting to lookup tables. A key contribution is an automated "cheating detection" mechanism, validated against human evaluation, to flag and penalize models that use memorized sequence values instead of generating a true algorithm. The study benchmarks a wide array of 21 leading LLMs, including reasoning-specialized models (like OpenAI's o-series and Google's Gemini 2.5-pro) and general-purpose models. The results show that reasoning-focused models significantly outperform their non-reasoning counterparts, especially on easy tasks. However, all models struggle significantly with the "hard" sequences, indicating that complex algorithmic reasoning remains a major challenge for the current state-of-the-art.

Strengths
- There reviewers find the benchmark design creative, especially the mechanism detecting "cheating" by the models.
- Comprehensive eval covering a variety of models.
- Public dataset, extensible for continual updates as OEIS grows.

Weaknesses
- difficulty  categorization is largely based on heuristics and may not fully reflect true difficulty.
- Cheating detection imperfect (~95% agreement with human judgments).
- Absence of prompt engineering for each model

The rebuttal partially addressed the reviewers' concern on the cheating detection. Overall, the reviewer find this submission a solid contribution that can be impactful in future research.